# Supramolecular and Macromolecular Matrix Nanocarriers for Drug Delivery in Inflammation-Associated Skin Diseases

**DOI:** 10.3390/pharmaceutics12121224

**Published:** 2020-12-17

**Authors:** Ranime Jebbawi, Séverine Fruchon, Cédric-Olivier Turrin, Muriel Blanzat, Rémy Poupot

**Affiliations:** 1INSERM, U1043, CNRS, U5282, UPS, Centre de Physiopathologie de Toulouse-Purpan, Université de Toulouse, F-31300 Toulouse, France; ranime.jebbawi@inserm.fr (R.J.); severine.fruchon@inserm.fr (S.F.); 2CNRS, UMR 5623, UPS, Laboratoire des Interactions Moléculaires et Réactivité Chimique et Photochimique, IMRCP, 118 Route de Narbonne, Université de Toulouse, CEDEX 9, F-31062 Toulouse, France; blanzat@chimie.ups-tlse.fr; 3CNRS, UPR 8241, Laboratoire de Chimie de Coordination, 205 Route de Narbonne, BP 44099, CEDEX 4, F-31077 Toulouse, France; cedric-olivier.turrin@lcc-toulouse.fr; 4LCC-CNRS, Université de Toulouse, CNRS, 31400 Toulouse, France

**Keywords:** atopic dermatitis, dendrimer, drug delivery, nanocarrier, psoriasis, skin inflammation, dermal delivery

## Abstract

Skin is our biggest organ. It interfaces our body with its environment. It is an efficient barrier to control the loss of water, the regulation of temperature, and infections by skin-resident and environmental pathogens. The barrier function of the skin is played by the stratum corneum (SC). It is a lipid barrier associating corneocytes (the terminally differentiated keratinocytes) and multilamellar lipid bilayers. This intricate association constitutes a very cohesive system, fully adapted to its role. One consequence of this efficient organization is the virtual impossibility for active pharmaceutical ingredients (API) to cross the SC to reach the inner layers of the skin after topical deposition. There are several ways to help a drug to cross the SC. Physical methods and chemical enhancers of permeation are a possibility. These are invasive and irritating methods. Vectorization of the drugs through nanocarriers is another way to circumvent the SC. This mini-review focuses on supramolecular and macromolecular matrices designed and implemented for skin permeation, excluding vesicular nanocarriers. Examples highlight the entrapment of anti-inflammatory API to treat inflammatory disorders of the skin.

## 1. Introduction

The skin is the largest organ in humans. For an average adult, its surface area is between 1.5 and 2 square meters, and its weight is between 4 and 5 kg (i.e., between 5% to 10% of the total body weight). The obvious role of the skin is to form a barrier between the “inside” and the “outside” [1]. As such, the skin prevents from pathogenic infections and fends off physical and chemical assaults. It also regulates the loss of solutes and water, insulation, and thereby the temperature of the body. Finally, it has a role in sensation, and in the synthesis of vitamin D.

To fulfill its physiological functions, the skin is composed of three main tissues, from the outside to the inside: epidermis, dermis, and hypodermis (Figure 1) [2]. The total thickness of the skin depends on its localization. It varies from 0.5 mm for the skin of the eyelids to 4–5 mm for the skin of the palms and the soles. The thickness of the skin also varies with gender and age. Epidermis is the outermost layer of the skin. It is a stratified, squamous, and keratinized epithelium composed of stacked strata, from the outside to the inside: stratum corneum (SC), stratum granulosum, stratum spinosum, and stratum basale, superimposed on the underlying lamina basale [2]. The average thickness of the epidermis varies from 50 to 150 µm. Its main cellular components are keratinocytes (90%). All along the life, these cells undergo a well-defined program of differentiation from the stratum basale to the SC, changing their shape, composition, and function. The duration of the differentiation of keratinocytes is about three weeks [3]. Along their differentiation, they will produce more and more keratin, and, eventually, they will lose their nucleus, release their cytoplasmic content, and undergo a death program. This is the keratinization, or cornification, process that leads to a waterproof epidermal barrier that will protect the organism from external assaults. When they are terminally differentiated, keratinocytes are called corneocytes, and they compose most of the SC. Besides keratinocytes, the epidermis encompasses Merkel cells (neuro-epithelial cells), Langherans cells (immunological cells), and melanocytes (responsible for the production of the melanin pigment). The epidermis is devoid of blood vessel. Underneath the epidermis lies the dermis of which cellular components are fibroblasts and immune cells, namely macrophages and mast cells [2]. The dermis is also composed of both a fibrillary matrix (collagen and elastin) and an extrafibrillar, gel-like, matrix (glycosaminoglycans, proteoglycans, and glycoproteins). Epidermis and dermis are tightly connected through the basal membrane. The dermis is vascularized and both nurtures and clears the epidermis. It is also innervated, providing the senses of touch and heat. The dermis is divided in two sublayers: the papillary region, adjacent to the epidermis, and the reticular region that is a deep and thicker area. The papillary region is named after the finger-like projections (papillae) extending towards the epidermis, interdigitating and, therefore, reinforcing the connection of the two tissues and increasing the surface for exchanges. The reticular region contains the hair follicles, and the sweat, sebaceous, and apocrine glands. The innermost layer is the subcutaneous hypodermis that is mainly composed of adipose cells. It is involved in the regulation of temperature and in the physical protection of the body [2]. It also connects the skin to the deeper tissues: bones and muscles.

Facing the outside world, the skin can be the site of physical, chemical, and infectious damages. Moreover, similar to any other organs, more or less severe dysfunctions and diseases are initiated from the inside. Therefore, a lot of different medical conditions can affect the skin. They are referred to as dermatoses. Any of the three tissues constituting the skin can be involved. Many of the most common dermatoses are related to inflammatory disorders. Among the latter, atopic dermatitis (AD) [4] and psoriasis [5] have the most important prevalence, even though it is difficult to get accurate numbers, especially for atopic dermatitis. It is generally admitted that around 20% of children and 10% of adults are concerned with AD in high-income countries [4]. The prevalence of psoriasis fluctuates from one country to another (from less than 1% to above 11%) [6], with more than 120 million patients affected worldwide. These diseases are mainly driven by inflammatory T cells infiltrating the dermis, also accompanied by polynuclear neutrophils, dendritic cells, and monocytes/macrophages [7]. Moreover, some immuno-competent resident cells of the skin will fuel and maintain inflammation (Langherans cells and keratinocytes).

Clearly, an advantageous route to achieve dermal delivery of a drug to the diseased area of the skin is the topical route of administration. It has many advantages: a direct targeting of the site of action, and a site-restricted distribution of the active pharmaceutical ingredient (API). Nevertheless, the barrier function of the skin strongly restricts the penetration of the API. Even under skin conditions in which the barrier function is impaired, it is highly challenging to get an API to pass through the outermost layer of the skin: the SC. The SC is actually an effective lipid barrier in comparison with other lipid barriers. For instance, skin and intestinal permeability properties are both based on lipids, but the intestinal epithelial cell monolayer is several orders of magnitude more permeable compared to the SC, which contains around 100 multilamellar lipid bilayers filling its intercorneocyte extracellular spaces. Several physicochemical parameters influence the capability of an API to permeate the SC, and the epidermis: size and molecular weight, lipophilicity, hydrogen-bonding groups, solubility, ionization of drug molecule, stereochemistry, and steric interaction [8,9]. In this regard, it has been shown that successful transdermal drugs are limited by parameter thresholds that are more restrictive than the Lipinsky’s Rule of Five [10], which is somehow quite restrictive and not adapted to encompass the reality of modern drug development in its complex heterogeneity. Nevertheless, the fact that the passive skin permeation is related to a more drastic lower limit on log P reflects the difficulty to cross SC. In this context, API scarcely gather the physicochemical features enabling the crossing of the SC. Therefore, drug delivery systems are needed to help with skin permeation and partition when passive permeation is not possible. A drug delivery system is a technology that enables an optimized therapeutic effect of an API through a precise control of its movement in the body or in a particular part thereof [11]. Over the past decades, nanotechnology has become pivotal in the domain of drug delivery, as in many others [12]. Different types of nanocarriers have been designed and implemented, for both dermal (when a local therapeutic effect is sought) and transdermal (when a systemic therapeutic effect is sought) drug delivery [13]. Nanocarriers are colloidal systems that are generally less than 500 nm in size [14]. They differ in structure and composition and can be classified into two large families: matrix nanocarriers and vesicular nanocarriers [15]. In this review, we focus on the development and use of supramolecular and macromolecular matrix nanocarriers for the dermal delivery of drugs in inflammation-associated diseases of the skin. Matrix systems are three-dimensional networks formed by lipids, surfactants, polymers, or dendrimers in which the API is trapped [15]. Herein, supramolecular matrix nanocarriers refer to colloidal nanoparticles made of surfactants (emulsions), lipids (SLN and NLC), and polymers (nanospheres), whereas macromolecular matrix nanocarriers are strictly reduced to unimolecular systems based on dendrimers.

## 2. Supramolecular Matrix Nanocarriers

### 2.1. Microemulsions and Nanoemulsions

Emulsions are heterogeneous systems composed of two immiscible liquids mixed together, water and oil, one of which is the dispersed phase, represented by thin droplets, and the other is the continuous phase. Droplets are stabilized thanks to surfactants, often in combination with co-surfactants. Emulsions are generally classified as: water-in-oil (W/O) emulsion, where the continuous phase is lipophilic and the dispersed phase is hydrophilic, and conversely for the oil-in-water (O/W) emulsion. Depending on the composition of the emulsion (selection of oil, surfactant, and co-surfactant) and the manufacturing process, the size of the droplets may vary and gives rise to different types of emulsions: whitish conventional emulsions (or macroemulsions) (size ranging from 1 to 10 μm), translucent microemulsions or nanoemulsions (size less than 100 or 200 nm), and multiple emulsions (W/O/W or O/W/O) [16]. In particular, according to Danielsson and Lindman, “microemulsion are a system of water, oil, and amphiphile which is a single optically isotropic and thermodynamically stable liquid solution” [17]. Microemulsions and nanoemulsions are macroscopically similar systems but are different types of colloidal dispersions as microemulsions are thermodynamically stable, unlike nanoemulsions [18]. Microemulsions are easy to prepare, as they form spontaneously, and inexpensive.

Several studies have shown the value of microemulsions in improving the skin penetration of hydrophilic (W/O emulsion) and lipophilic (O/W emulsion) molecules [19]. One of the mechanisms by which microemulsions promote skin penetration is the fluidization of the lipid bilayer of the SC by disrupting the organization of intercellular lipids. This is due to the enhancers of penetration, such as surfactants and lipids, which are used in the formulation. These systems can also promote skin absorption by enhancing the solubility of the API in the skin [20].

Several reviews deal with the topic of the delivery of anti-psoriatic drugs formulated in nanoemulsions [21]. Recent examples of the formulation of anti-inflammatory drugs in microemulsions for dermal delivery include the steroid betamethasone dipropionate [22], used to treat psoriasis, and a mix of vitamins A and E to treat acute skin inflammation in a mouse model [23]. Different nanocarrier-based formulations, including microemulsions, of anti-inflammatory retinoids used for the management of acne vulgaris are recently reviewed [24]. However, the large amounts of lipids and surfactants present in microemulsions can lead to toxic effects that hamper their use [25].

### 2.2. Nanospheres

Nanospheres are solid colloidal particles, with a diameter between 100 and 200 nm, composed of polymers. They entrap the API in their polymer matrix, but it is also possible that the API is adsorbed on the surface of the matrix [26]. The polymers used for the formation of nanospheres are often biocompatible, and can be classified according to their origin in: (i) natural polymers (or biopolymers), such as chitosan, albumin, alginate, and gelatin; (ii) non-degradable synthetic polymers, such as polyacrylates; and (iii) biodegradable synthetic polymers, such as polymers derived from tyrosine, polylactic acids, polyglycolic acids, poly(lactic-*co*-glycolic acid) copolymers, and poly(ε-caprolactones) [27]. Studies have shown that these rigid polymeric nanoparticles do not penetrate beyond the superficial SC and exhibit some affinity for the hair follicles through which they can penetrate. Nanospheres may therefore prove useful as reservoirs of API at the surface of the skin, controlling the release thereof in the deeper layers of the skin [28,29].

Nanospheres composed of amphiphilic polymers derived from tyrosine, called tyrospheres, have shown their effectiveness for the topical delivery of hydrophobic drugs by improving their solubility, and therefore their skin penetration. These drugs include paclitaxel [30], cholecalciferol [31], and cyclosporine [32], indicated for the treatment of psoriasis, and adapalene [33], indicated for the treatment of acne vulgaris.

It is important to remember that one of the necessary conditions for placing a pharmaceutical product on the market is the existence of an appropriate and accurately controlled method for large-scale production. Thus, the lack of a simple and cost-effective method for the production of nanospheres on a large scale hinders their industrial development. In addition, some synthetic polymers, especially polyesters, are known to be phagocytosed by macrophages that will release degradation products with potential cytotoxic effects [34]. On the other hand, natural polymers are less toxic, but batch-to-batch variability makes it difficult to obtain pharmaceutical products with consistent characteristics [27].

### 2.3. Solid Lipid Nanoparticles (SLNs)

Designed in the early 1990s, SLNs are particles of solid lipids dispersed in a liquid medium and stabilized by surfactants [35]. Their size varies between 50 and 1000 nm [36]. SLNs have many advantages such as biocompatibility, afford protection of the API against chemical degradation in their solid matrix, and control of drug release [35]. Unlike nanospheres, SLNs have the advantage of being able to be produced in a simple way on a large scale [37]. Moreover, due to the adhesive properties of small particles, SLNs form a hydrophobic film on the surface of the skin which has an occlusive effect [35]. Occlusion limits the loss of water, and therefore promotes the hydration of the skin. Subsequently, it will lead to a widening of the intercorneocyte spaces, and therefore will increase the mobility of the intercellular lipid chains. Overall, this potentiates the penetration of the API into the SC [38]. Additionally, SLN lipids may fuse with lipids of the SC, thereby increasing the permeation of the API. Jensen et al. demonstrated that the solubility of the API in the lipids used in SLNs and the size of the resulting nanoparticles can also affect the penetration profile of the API in the skin. They evaluated in vitro the penetration profile of the steroid betamethasone entrapped into SLNs differing in their lipid composition. They tested SLNs composed of glyceryl-distearate, glyceryl-tripalmitate, and cetyl palmitate. The permeation effect of the three SLNs was compared to the effect of a conventional ointment (paraffin) that also has occlusive effect and is suitable for the delivery of lipophilic drugs into the skin. This study has shown that the permeation of the API through the skin is reduced when formulated in SLNs compared to the ointment, but a greater amount of the API is retained at the SC when administered with SLN formulations. Among the three different SLNs, the one composed of glyceryl-distearate enabled the persistence of higher amounts of betamethasone at the SC. This is probably due to the better solubility of the API in this type of SLNs and the smaller size of these SLNs compared to those composed of other lipids [39]. In conclusion, similar to solid polymeric nanoparticles such as nanospheres, SLNs remain at the SC, creating a reservoir of API which prolongs its permeation in the skin. Another anti-psoriatic drug, mometasone furoate, has been successfully entrapped in SLNs, enhancing its penetration in the skin in an in vitro model [40].

However, SLNs have a limited capacity to carry API, and the release thereof is not always controlled due to the high crystallinity of their matrix. These factors have limited the application of SLNs in the vectorization of API ingredients [38] and have led to the development of nanostructured lipid nanoparticles (NLC).

### 2.4. Nanostructured Lipid Carriers (NLCs)

NLCs are the second generation of lipid nanoparticles. Their lipid matrix is formed by a mixture of structurally very different lipids, i.e. solid lipids with long chain fatty acids and liquid lipids (oils) with short chain fatty acids. NLCs have various advantages similar to those of SLNs, such as the use of biocompatible lipids, large-scale production, and the protection of drugs from degradation [41]. Contrary to SLN that are composed of solid lipids only, the introduction of liquid lipids into the matrix leads to a reduction in the order of lipids, and therefore to an imperfect crystal structure, which may provide more space for inclusion of API [42]. This makes it possible to obtain both higher encapsulation rates and better stability over time with NLCs Interestingly, the same anti-psoriatic drug (mometasone furoate) entrapped in SLNs has been entrapped in NLCs [43]. Contrary to what one may have expected, the entrapment rates are approximately the same (56% in SLNs vs. 60% in NLCs). The skin deposition of the API is 2.67- and 2.5-fold higher with SLNs and NLCs respectively, when compared to the marketed cream formulation of the API. Contrary to the SLN formulation, the NLC one has been tested in an in vivo mouse model of psoriasis and has shown therapeutic benefit [43]. Viegas et al. recently showed the potential of NLCs for the combined delivery of siRNA directed against TNF and of the anti-psoriatic drug tacrolimus. The anti-psoriatic effect of this combination co-delivered with NLCs has been validated in vivo [44]. Despite their effectiveness to entrap hydrophobic API, the encapsulation of hydrophilic API in NLCs remains limited.

When available in the articles cited in this section, permeability coefficients (K_p_) and steady state flux (J_SS_) are compiled in Table 1.

## 3. Macromolecular Matrix Nanocarriers: Dendrimers

Dendrimers are hyperbranched, multivalent, and polyfunctional macromolecules. They are designed and synthesized from a central core, to which are linked one or several series of branched monomers. Each monomer is ended by a divergence point that enables the branching of an additional generation of branched monomers. This is the so-called dendritic growth that leads to “tree-like” macromolecules. The synthesis ends with the addition of surface functional groups on the last-added series of branched monomers (Figure 2) [50]. The total number of series of branched layers determines the generation of the dendrimer. Generally, this highly controlled, stepwise synthesis affords dendrimers with perfectly defined size and structure, especially for low generation dendrimers. Therefore, dendrimers are a promising alternative to poorly defined nanoparticles, such as linear polymer or metallic ones, for biomedical applications. Several other intrinsic features of dendrimers, such as their high degree of functionality, their globular shape, and nanometric size, their multivalency, are responsible for the long-lasting interest of the scientific community to develop dendrimer-based therapeutic approaches [50,51,52]. Interestingly, in the very first report about the molecules that are considered the ancestors of dendrimers, the presence of cavities at the inner part of the structure is highlighted [53]. Years later, in biomedical applications, dendrimers are used as nanocarriers for imaging agents, nucleic acids, and drugs. Thanks to the high degree of functionality of dendrimers, molecules can be entrapped in the cavities of dendrimers, bound via electrostatic interactions at their surface, or covalently conjugated with their surface groups [54]. Although numerous studies have shown that dendrimers enhance the skin permeation of API, the basic mechanisms underpinning this property are poorly deciphered. It seems that hydrogen-bonding and electrostatic interactions of dendrimers with the polar heads of phospholipids of the multilamellar bilayers of the SC could account for the enhancement of permeation [45,55]. It has been shown that dendrimers induce the formation of holes and defaults of nanometric size in both lipid bilayers and cell membranes [56,57]. Moreover, dendrimers also enhance the solubility and stability of API, increasing thereby their skin permeation and partition [58].

The first report mentioning vectorization of an anti-inflammatory API with dendrimers can be traced back to 2003 [46]. In this pioneering study, the authors evaluated the efficacy of three poly(amidoamine) (PAMAM) dendrimers for transdermal drug delivery. PAMAM dendrimers have a diaminoethane core and amidoamine branched monomers. In the abovementioned study, PAMAM dendrimers of the fourth generation were ended by cationic –NH_3_^+^, anionic –COO^−^, or neutral –OH surface groups. They showed that both cationic and neutral PAMAM dendrimers enhance the permeation of the non-steroidal indomethacin, with an advantage for the cationic nanocarrier. It was concluded that the enhancement of indomethacin permeation was due to a higher cargo of the API within the cationic dendrimer. Indeed, this dendrimer can interact with the acidic group of indomethacin at its surface and entrap the drug in its hydrophobic cavities. On the contrary, the anionic dendrimer has only the possibility to entrap the API. Later on, others have shown that the molecular size (related to the generation of the dendrimer) is also a key parameter for skin permeation [45]. It has been shown that cationic PAMAM dendrimers of the second generation can permeate deeper in the epidermis than dendrimers of the fourth generation [47,59,60]. The advantage of cationic dendrimers to permeate the skin could be linked to their ability to interact with negatively charged lipid bilayers of the SC [61].

Dithranol is an anti-psoriatic drug that is sensitive to light, and that induces irritation, burning sensation, and staining. Therefore, vectorization is a smart strategy to overcome the troublesome and inconvenient topical application of this API. A PolyPropylene Imine (PPI) dendrimer of the fifth generation harboring –NH_3_^+^ functions at its surface has been used to more efficiently deliver dithranol [48]. In this study, the improvement in the penetration of API is not due to the penetration of the dendrimer into the skin, but rather to a better aqueous solubility and an increase of the partition coefficient in the SC by the fifth generation PPI dendrimer, therefore improving the diffusion of the API in the skin. In the referenced study, the loaded nanocarrier has been evaluated for tolerance and toxicity, but not yet tested in an animal model of psoriasis.

Another possibility to control inflammation is to neutralize pivotal inflammatory cytokines, such as Tumor Necrosis Factor (TNF). This had been tested by topical delivery of a silencing RNA against the TNF transcript. This siRNA has been loaded on to a PAMAM cationic dendrimer, and the nanosystem has been evaluated for anti-psoriatic therapeutic effect in a mouse model of psoriasis. This assay has shown an improvement of both phenotypic and histopathological features of the disease, with a reduction of the level of inflammatory cytokines TNF as expected, but also interleukin (IL)-6 and IL-17 [62].

To end, we mention a hybrid dendrimer-based nanocarrier designed for topical delivery of the anti-inflammatory diclofenac sodium [49]. This system conjugates a poly(propyletherimine) (PETIM) dendrons of the first and second generations to the unsaturated oleic acid. The latter has been chosen because it was shown formerly that it is an enhancer of the permeation of diclofenac sodium [63]. These innovative nanocarriers show significant increase in the deposition of the API in the skin as compared to the API alone or combined with oleic acid.

When available in the articles cited in this section, permeability coefficients (K_p_) and steady state flux (J_SS_) are compiled in Table 1.

## 4. Conclusions

Among the number of existing systems, we focus our review on the most advanced matrix nanocarriers used in dermal delivery. However, vesicular nanocarriers are also developed to achieve dermal delivery of a drug. Vesicles are colloidal nanocarriers containing a liquid core, separated from the outside by a membrane made of phospholipids, surfactants, or polymers [64]. Liposomes and their derivatives ethosomes and transfersomes [65] (and the more recent flavosomes [66] and invasomes [67], among others) based on lipids, niosomes [68] and catanionic vesicles [69] made of surfactants, and nanocapsules [70] and polymersomes made of polymers are vesicular nanocarriers. Depending on the nature and physicochemical properties of the vesicle membrane, vesicular nanocarriers can also succeed in improving the dermal penetration of an API [71,72].

Supramolecular matrix nanocarriers have all proved to promote skin penetration of the API by improving its solubility. Microemulsions also enhance API skin penetration by fluidizing the lipid bilayers of the SC, while lipid nanoparticles (SLN and NLC) form an occlusive film on the surface of the skin that widens the lipid intercellular spaces. Nevertheless, most of these supramolecular matrix systems only improve the penetration of the drug in the outermost layers of the skin and do not deeply penetrate in the skin. However, they have been shown to be able to improve the drug efficacy as they play the role of API reservoirs at the skin surface. Recent reviews show the benefits of variable topical nanocarriers (polymer-based nanoparticles, lipid nanocarriers, inorganic nanoparticles, and nanoparticles incorporated in hydrogels) in improving the therapeutic efficacy of conventional anti-AD [73] and anti-psoriatic [74] drugs in mouse models and, sometimes, in clinical studies. Over the past decades, nanotechnology has become pivotal in the domain of drug delivery, as in many others, and many systems have reached clinical trials, especially for systemic delivery [75,76]. Nevertheless, several challenges are still to be tackled for the clinical translation of nanocarriers, for both systemic and topical uses, such as toxicity, immunological and environmental issues [12], physicochemical design [76], and scale-up of production processes [73].

Cationic dendrimers, especially PAMAM and PPI ones, have shown promising nanocarrier properties for the enhancement of skin permeation of API. Nevertheless, it has to be mentioned that cationic dendrimers, including PAMAM and poly(phosphorhydrazone) (PPH) ones, are cytotoxic [77,78], including on rat skin model for PAMAM dendrimers [79]. It is acknowledged that these cytotoxic effects take place because of the strong interactions between the numerous cationic groups displayed at the surface of these dendrimers and polyanionic structures of living cells: lipid bilayers and nucleic acids. On the contrary, a PPH dendrimer of the first generation capped with anionic surface groups has shown promising safety, including immuno-safety, and tolerance in different animal models [80,81]. Of note, the same dendrimer has intrinsic anti-inflammatory properties which have shown therapeutic efficacy in animal models of inflammatory disorders [82,83,84], including the imiquimod-induced psoriasis mouse model [85]. PAMAM dendrimers that are commonly used as macromolecular nanocarriers also have intrinsic anti-inflammatory effects [51,86]. Finally, it has to be mentioned that the first ever marketed preparation of dendrimers (Vivagel^®^; Starpharma Holdings Ltd., Melbourne, Australia) is a topical polyanionic dendrimer, intended for the treatment of sexually transmitted infections for women.

## Figures and Tables

**Figure 1 pharmaceutics-12-01224-f001:**
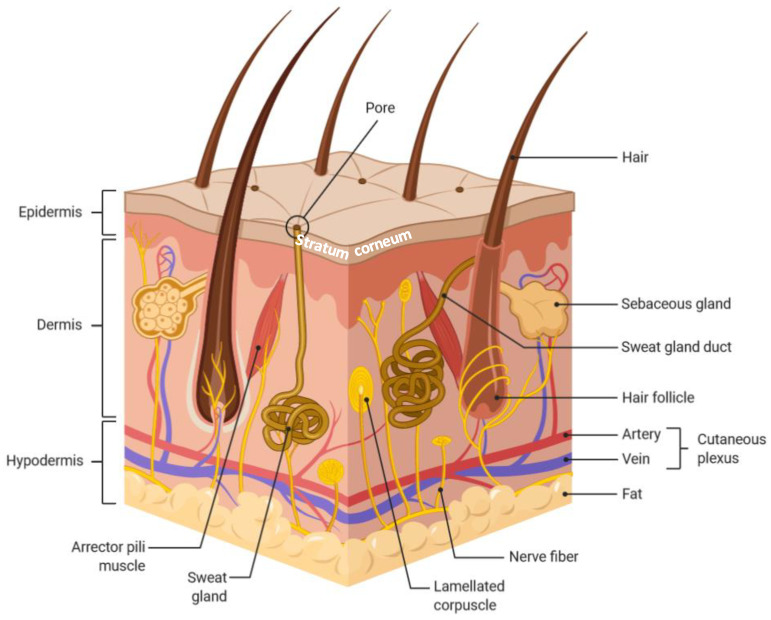
Schematic representation of the three layers of the skin with the skin appendages, the vasculature, and the nerves (created with BioRender.com).

**Figure 2 pharmaceutics-12-01224-f002:**
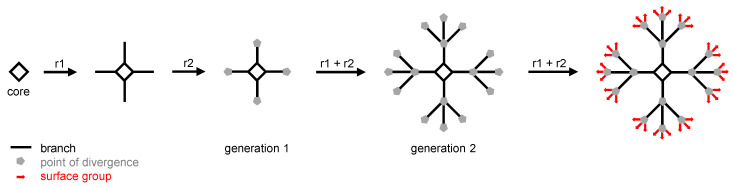
Schematic synthesis of a Generation 2 (two series of branches) dendrimer using a tetravalent core (i.e., four branches in the first series) and trivalent points of divergence (i.e., 12 branches in the second series and 36 surface groups). r1 and r2 are the reactions which are iterated to obtain the final dendrimer.

**Table 1 pharmaceutics-12-01224-t001:** Compilation of the studies cited in the current review and presenting permeability coefficient and steady state flux measurements with enhancement of these parameters using different types of nanocarriers.

API	Nanocarriers	Permeability Coefficient, K_p_ (cm/h)	Steady State Flux, J_SS_ (µg/cm^2^/h)	Reference
*minoxidil*	nanoemulsion	Nd ^1^	1.9 ^2^/18.1	[20]
*betamethasone dipropionate*	microemulsion	nd	3.95	[22]
	NLC	nd	1.53	
	SLN	nd	0.79	
*adapalene*	tyrospheres	nd	1.4	[33]
*mometasone furoate*	drug-loaded gel	9.82 × 10^−3^	3.27	[43]
	NLC dispersion	5.76 × 10^−3^	1.93	
	NLC-based hydrogel	4.47 × 10^−3^	1.49	
	marketed formulation	10.51 × 10^−3^	3.50	
*5-fluorouracyl*	PAMAM dendrimers	1.86	67.0	[45] dendrimers are used as pretreatment
	G2-cationic	5.39	194.4	
	G4-cationic 0.1/1/10 mM	3.48/4.65/6.54	125.5/167.5/235.6	
	G6-cationic	2.67	96.4	
	G4-neutral	3.62	130.5	
	G3.5-anionic	2.79	100.5	
*indomethacin*	PAMAM dendrimers:	61.2 × 10^−3^	1.53	[46]
	G4.5-anionic	15.0 × 10^−3^	1.83	
	G4-neutral	6.7 × 10^−3^	2.17	
	G4-cationic	10.1 × 10^−3^	3.77	
*salicylic acid*	PAMAM dendrimers	nd	38.5	[47] dendrimers are used as pretreatment
	G2-cationic (1 and 10 mM)	nd	38.1/66.3	
	G3-cationic	nd	34.3	
*dithranol*	PPI dendrimers	nd	2.72	[48]
	G5-cationic	nd	11.61	
*diclofenac sodium*	Oleodendrimers (PETIM dend. + oleic acid)	0.58 × 10^−4^	0.87	[49]
	oleic acid as enhancer	1.09 × 10^−4^	1.63	
	oleodendrimer E1E	1.94 × 10^−4^	2.90	
	oleodendrimer E2E	1.97 × 10^−4^	2.95	
	oleodendrimer A1E	1.17 × 10^−4^	1.76	
	oleodendrimer A2E	1.50 × 10^−4^	2.24	

^1^ Not determined; ^2^ values in italic are for API alone or control.

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
