# Peer review of "Supramolecular and Macromolecular Matrix Nanocarriers for Drug Delivery in Inflammation-Associated Skin Diseases"

_pharmaceutics, 2020, doi:10.3390/pharmaceutics12121224_

Round 1
Reviewer 1 Report
1. Cite additional recent references
2. Corelate the reported work w.r.t clinical significance in the field (in the conclusion section)
Author Response
Comments and Suggestions for Authors
- Cite additional recent references
2. Correlate the reported work w.r.t clinical significance in the field (in the conclusion section)
Response: a paragraph commenting on clinical translation of nanocarriers and toxicity issues thereof has been added in the conclusion section. Several recent references have been added in this paragraph.
Reviewer 2 Report
This is a very good review that carries a good idea of classifying the matrix-based systems for skin delivery.
I just wonder if this manuscript is considered a review or a mini review?
Please elaborate the difference between supramolecular and macromolecular in the introduction.
I think the polymeric nanospheres should be transferred to the macromolecular section.
Please add a table stating the benefits of the usage of each type of the used nanocarriers in skin delivery, comparing parameters such as the permeability coefficient or steady state flux and favorably for the same drugs (if present).
Author Response
Comments and Suggestions for Authors
This is a very good review that carries a good idea of classifying the matrix-based systems for skin delivery.
Thank you very much for your valuable comment. Please find below our responses to your comments and suggestions.
I just wonder if this manuscript is considered a review or a mini review?
Response: we will comply with the Editor’s choice.
Please elaborate the difference between supramolecular and macromolecular in the introduction.
Response: a sentence has been added at the end of the introduction.
I think the polymeric nanospheres should be transferred to the macromolecular section.
Response: nanospheres are supramolecular systems of macromolecules. Therefore, it can be included in both sections. We propose to leave this paragraph in the supramolecular section, but we will comply with the Editor’s decision.
Please add a table stating the benefits of the usage of each type of the used nanocarriers in skin delivery, comparing parameters such as the permeability coefficient or steady state flux and favorably for the same drugs (if present).
Response: we have added a table as requested. This table is based on the cited references that present data with permeability coefficient and/or steady state flux measurements.
Reviewer 3 Report
Jebbawi and coworkers proposed to review the use of matrix nanovectors for the treatment of inflammation-associated skin diseases. This class includes different and important skin diseases that severely affect the quality of life of the patients and are the object of intense research, since the cure for these pathologies is not available still today. The use of nanovectors as means to improve the treatment of inflammation based skin diseases has been the topic of different reviews present in the literature, most of them also recent. Just to provide some examples:
- Dadwal A, Mishra N, Narang RK. Novel Topical Nanocarriers for Treatment of Psoriasis: An Overview. Curr Pharm Des. 2018;24(33):3934-3950. doi: 10.2174/1381612824666181102151507. PMID: 30387390.
- Estefânia Vangelie Ramos Campos, Patrícia Luiza De Freitas Proença, Lorena Doretto-Silva, Vinicius Andrade-Oliveira, Leonardo Fernandes Fraceto & Daniele Ribeiro de Araujo (2020) Trends in nanoformulations for atopic dermatitis treatment, Expert Opinion on Drug Delivery, 17:11, 1615-1630, DOI: 1080/17425247.2020.1813107
- Gupta M, Agrawal U, Vyas SP. Nanocarrier-based topical drug delivery for the treatment of skin diseases. Expert Opin Drug Deliv. 2012 Jul;9(7):783-804. doi: 10.1517/17425247.2012.686490. Epub 2012 May 5. PMID: 22559240.
- Abdel-Mottaleb MM, Try C, Pellequer Y, Lamprecht A. Nanomedicine strategies for targeting skin inflammation. Nanomedicine (Lond). 2014 Aug;9(11):1727-43. doi: 10.2217/nnm.14.74. PMID: 25321172.
Beside the lack of mention for these reviews in the manuscript, the present paper does not provide new insight into the use of nanovectors for skin inflammation but instead the discussion is poor as well as the presented data. In other words, in my opinion this paper does not add novelty to the existing literature and does not provide food for thought for researchers.
I would like also to add with reference to line 129 of the text, that microemulsions and nanoemulsions are not the same system.
Author Response
Comments and Suggestions for Authors
Jebbawi and coworkers proposed to review the use of matrix nanovectors for the treatment of inflammation-associated skin diseases. This class includes different and important skin diseases that severely affect the quality of life of the patients and are the object of intense research, since the cure for these pathologies is not available still today. The use of nanovectors as means to improve the treatment of inflammation based skin diseases has been the topic of different reviews present in the literature, most of them also recent. Just to provide some examples:
- Dadwal A, Mishra N, Narang RK. Novel Topical Nanocarriers for Treatment of Psoriasis: An Overview. Curr Pharm Des. 2018;24(33):3934-3950. doi: 10.2174/1381612824666181102151507. PMID: 30387390.
- Estefânia Vangelie Ramos Campos, Patrícia Luiza De Freitas Proença, Lorena Doretto-Silva, Vinicius Andrade-Oliveira, Leonardo Fernandes Fraceto & Daniele Ribeiro de Araujo (2020) Trends in nanoformulations for atopic dermatitis treatment, Expert Opinion on Drug Delivery, 17:11, 1615-1630, DOI: 1080/17425247.2020.1813107
- Gupta M, Agrawal U, Vyas SP. Nanocarrier-based topical drug delivery for the treatment of skin diseases. Expert Opin Drug Deliv.2012 Jul;9(7):783-804. doi: 10.1517/17425247.2012.686490. Epub 2012 May 5. PMID: 22559240.
- Abdel-Mottaleb MM, Try C, Pellequer Y, Lamprecht A. Nanomedicine strategies for targeting skin inflammation. Nanomedicine (Lond).2014 Aug;9(11):1727-43. doi: 10.2217/nnm.14.74. PMID: 25321172.
Response: the two most recent references proposed by the reviewer have been added in the conclusion section in a paragraph commenting on clinical translation of nanocarriers and toxicity issues thereof.
Beside the lack of mention for these reviews in the manuscript, the present paper does not provide new insight into the use of nanovectors for skin inflammation but instead the discussion is poor as well as the presented data. In other words, in my opinion this paper does not add novelty to the existing literature and does not provide food for thought for researchers.
I would like also to add with reference to line 129 of the text, that microemulsions and nanoemulsions are not the same system.
Response: we have added a sentence explaining that microemulsions and nanoemelusions are different systems: “Microemulsions and nanoroemulsions are macroscopically similar systems but are different types of colloidal dispersions as microemulsions are thermodynamically stable, unlike nanoemulsions”, with a reference (new ref 18).